# CyCADA: Cycle-Consistent Adversarial Domain Adaptation

## Abstract

Domain adaptation is critical for success in new, unseen environments. Adversarial adaptation models applied in feature spaces discover domain invariant representations, but are difficult to visualize and sometimes fail to capture pixel-level and low-level domain shifts. Recent work has shown that generative adversarial networks combined with cycle-consistency constraints are surprisingly effective at mapping images between domains, even without the use of aligned image pairs. We propose a novel discriminatively-trained Cycle-Consistent Adversarial Domain Adaptation model. CyCADA adapts representations at both the pixel-level and feature-level, enforces cycle-consistency while leveraging a task loss, and does not require aligned pairs. Our model can be applied in a variety of visual recognition and prediction settings. We show new state-of-the-art results across multiple adaptation tasks, including digit classification and semantic segmentation of road scenes demonstrating transfer from synthetic to real world domains.

## 1 Introduction

Deep neural networks excel at learning from large amounts of data, but can be poor at generalizing learned knowledge to new datasets or environments. Even a slight departure from a network's training domain can cause it to make spurious predictions and significantly hurt its performance (Tzeng et al., 2017). The visual domain shift from non-photorealistic synthetic data to real images presents an even more significant challenge. While we would like to train models on large amounts of synthetic data such as data collected from graphics game engines, such models fail to generalize to real-world imagery. For example, a state-of-the-art semantic segmentation model trained on synthetic dashcam data fails to segment the road in real images, and its overall per-pixel label accuracy drops from 93% (if trained on real imagery) to 54% (if trained only on synthetic data, see Table 5).

Feature-level unsupervised domain adaptation methods address this problem by aligning the features extracted from the network across the source (e.g. synthetic) and target (e.g. real) domains, without any labeled target samples. Alignment typically involves minimizing some measure of distance between the source and target feature distributions, such as maximum mean discrepancy (Long & Wang, 2015), correlation distance (Sun & Saenko, 2016), or adversarial discriminator accuracy (Ganin & Lempitsky, 2015; Tzeng et al., 2017). This class of techniques suffers from two main limitations. First, aligning marginal distributions does not enforce any semantic consistency, e.g. target features of a car may be mapped to source features of a bicycle. Second, alignment at higher levels of a deep representation can fail to model aspects of low-level appearance variance which are crucial for the end visual task.

Generative pixel-level domain adaptation models perform similar distribution alignment—not in feature space but rather in raw pixel space—translating source data to the "style" of a target domain. Recent methods can learn to translate images given only unsupervised data from both domains (Bousmalis et al., 2017b; Liu & Tuzel, 2016b; Shrivastava et al., 2017). The results are visually compelling, but such image-space models have only been shown to work for small image sizes and limited domain shifts. A more recent approach (Bousmalis et al., 2017a) was applied to larger (but still not high resolution) images, but in a controlled environment with visually simple images for robotic applications. Furthermore, they also do not necessarily preserve content: while the translated image may "look" like it came from the right domain, crucial semantic information may be lost. For

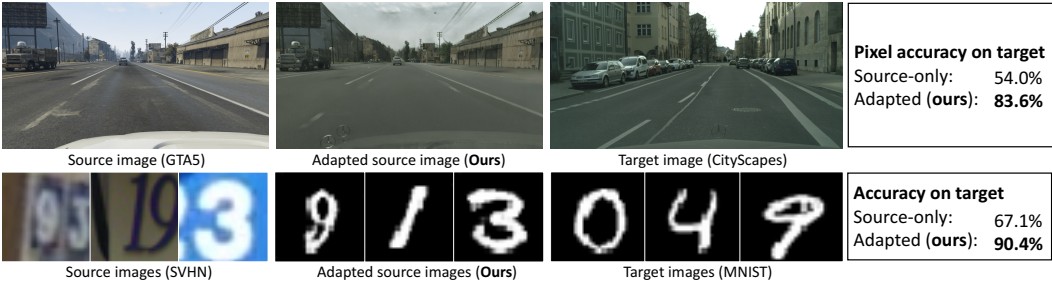

Figure 1: We propose CyCADA, an adversarial unsupervised adaptation algorithm which uses cycle and semantic consistency to perform adaptation at multiple levels in a deep network. Our model provides significant performance improvements over source model baselines.

|  | Pixel Loss | Feature Loss | Semantic Loss | Cycle Consistent |
|---|---|---|---|---|
| CycleGAN (Zhu et al., 2017) | ✓ |  |  | ✓ |
| Feature Adapt (Ganin & Lempitsky, 2015; Tzeng et al., 2017) |  | ✓ | ✓ |  |
| Pixel Adapt (Taigman et al., 2017a; Bousmalis et al., 2017b) | ✓ |  | ✓ |  |
| CyCADA | ✓ | ✓ | ✓ | ✓ |

Table 1: Our model, CyCADA, may use pixel, feature, and semantic information during adaptation while learning an invertible mapping through cycle consistency.

example, a model adapting from line-drawings to photos could learn to make a line-drawing of a cat look like a photo of a dog.

How can we encourage the model to preserve semantic information in the process of distribution alignment? In this paper, we explore a simple yet powerful idea: give an additional objective to the model to reconstruct the original data from the adapted version. Cycle-consistency was recently proposed in a cross-domain image generation GAN model, CycleGAN (Zhu et al., 2017), which showed transformative image-to-image generation results, but was agnostic to any particular task.

We propose Cycle-Consistent Adversarial Domain Adaptation (CyCADA), which adapts representations at both the pixel-level and feature-level while enforcing pixel and semantic consistency. We use a reconstruction (cycle-consistency) loss to enforce the cross-domain transformation to preserve pixel information and a semantic labeling loss to enforce semantic consistency. CyCADA unifies prior feature-level (Ganin & Lempitsky, 2015; Tzeng et al., 2017) and image-level (Liu & Tuzel, 2016b; Bousmalis et al., 2017b; Shrivastava et al., 2017) adversarial domain adaptation methods together with cycle-consistent image-to-image translation techniques (Zhu et al., 2017), as illustrated in Table 1. It is applicable across a range of deep architectures and/or representation levels, and has several advantages over existing unsupervised domain adaptation methods.

We apply our CyCADA model to the task of digit recognition across domains and the task of semantic segmentation of urban scenes across domains. Experiments show that our model achieves state of the art results on digit adaptation, cross-season adaptation in synthetic data, and on the challenging synthetic-to-real scenario. In the latter case, it improves per-pixel accuracy from 54% to 83%, nearly closing the gap to the target-trained model.

Our experiments confirm that domain adaptation can benefit greatly from cycle-consistent pixel transformations, and that this is especially important for pixel-level semantic segmentation with contemporary FCN architectures. We demonstrate that enforcing semantic consistency between input and stylized images prevents label flipping on the large shift between SVHN and MNIST (example, prevents a SVHN 9 from being mapped into an MNIST 2). Interestingly, on our semantic segmentation tasks (GTA to CityScapes) we did not observe label flipping to be a major source of error, even without the semantic consistency loss. Because of this, and due to memory constraints, we do not include this loss for the segmentation tasks. Further, we show that adaptation at both the pixel and representation level can offer complementary improvements with joint pixel-space and feature adaptation leading to the highest performing model for digit classification tasks.

## 2 RELATED WORK

The problem of visual domain adaptation was introduced along with a pairwise metric transform solution by Saenko et al. (2010) and was further popularized by the broad study of visual dataset bias (Torralba & Efros, 2011). Early deep adaptive works focused on feature space alignment through minimizing the distance between first or second order feature space statistics of the source and target (Tzeng et al., 2014; Long & Wang, 2015). These latent distribution alignment approaches were further improved through the use of domain adversarial objectives whereby a domain classifier is trained to distinguish between the source and target representations while the domain representation is learned so as to maximize the error of the domain classifier. The representation is optimized using the standard minimax objective (Ganin & Lempitsky, 2015), the symmetric confusion objective (Tzeng et al., 2015), or the inverted label objective (Tzeng et al., 2017). Each of these objectives is related to the literature on generative adversarial networks (Goodfellow et al., 2014) and follow-up work for improved training procedures for these networks (Salimans et al., 2016b; Arjovsky et al., 2017).

The feature-space adaptation methods described above focus on modifications to the discriminative representation space. In contrast, other recent methods have sought adaptation in the pixel-space using various generative approaches. One advantage of pixel-space adaptation, as we have shown, is that the result may be more human interpretable, since an image from one domain can now be visualized in a new domain. CoGANs (Liu & Tuzel, 2016b) jointly learn a source and target representation through explicit weight sharing of certain layers while each source and target has a unique generative adversarial objective. Ghifary et al. (2016) uses an additional reconstruction objective in the target domain to encourage alignment in the unsupervised adaptation setting.

In contrast, another approach is to directly convert the target image into a source style image (or visa versa), largely based on Generative Adversarial Networks (GANs) (Goodfellow et al., 2014). Researchers have successfully applied GANs to various applications such as image generation (Denton et al., 2015; Radford et al., 2015; Zhao et al., 2016), image editing (Zhu et al., 2016) and feature learning (Salimans et al., 2016a; Donahue et al., 2017). Recent work (Isola et al., 2016; Sangkloy et al., 2016; Karacan et al., 2016) adopt conditional GANs (Mirza & Osindero, 2014) for these image-to-image translation problems (Isola et al., 2016), but they require input-output image pairs for training, which is in general not available in domain adaptation problems.

There also exist lines of work where such training pairs are not given. Yoo et al. (2016) learns a source to target encoder-decoder along with a generative adversarial objective on the reconstruction which is is applied for predicting the clothing people are wearing. The Domain Transfer Network (Taigman et al., 2017b) trains a generator to transform a source image into a target image by enforcing consistency in the embedding space. Shrivastava et al. (2017) instead uses an L1 reconstruction loss to force the generated target images to be similar to their original source images.This works well for limited domain shifts where the domains are similar in pixel-space, but can be too limiting for settings with larger domain shifts. Bousmalis et al. (2017b) use a content similarity loss to ensure the generated target image is similar to the original source image; however, this requires prior knowledge about which parts of the image stay the same across domains (e.g. foreground). Our method does not require pre-defining what content is shared between domains and instead simply translates images back to their original domains while ensuring that they remain identical to their original versions. BiGAN (Donahue et al., 2017) and ALI (Dumoulin et al., 2016) take an approach of simultaneously learning the transformations between the pixel and the latent space. More recently, Cycle-consistent Adversarial Networks (CycleGAN) (Zhu et al., 2017) produced compelling image translation results such as generating photorealistic images from impressionism paintings or transforming horses into zebras at high resolution using the cycle-consistency loss. This loss was simultaneously proposed by Yi et al. (2017) and Kim et al. (2017) to great effect as well. Our motivation comes from such findings about the effectiveness of the cycle-consistency loss.

Few works have explicitly studied visual domain adaptation for the semantic segmentation task. Adaptation across weather conditions in simple road scenes was first studied by Levinkov & Fritz (2013). More recently, a convolutional domain adversarial based approached was proposed for more general drive cam scenes and for adaptation from simulated to real environments (Hoffman et al., 2016). Ros et al. (2016b) learns a multi-source model through concatenating all available labeled data and learning a single large model and then transfers to a sparsely labeled target domain through distillation (Hinton et al., 2015). Chen et al. (2017) use an adversarial objective to align both global

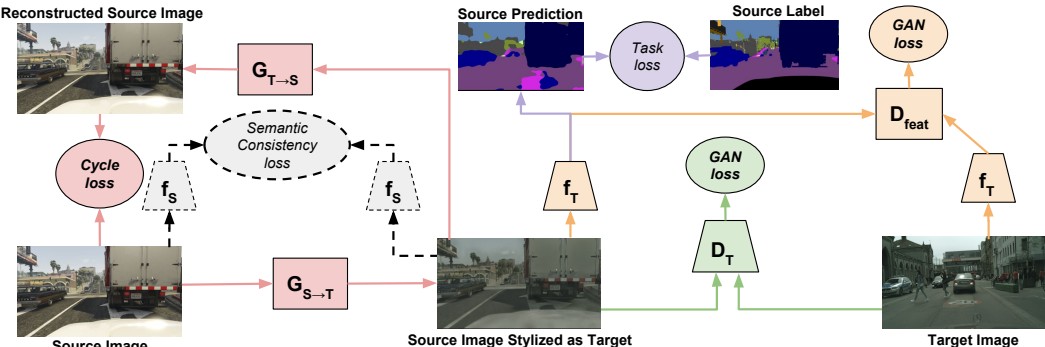

Figure 2: Cycle-consistent adversarial adaptation overview. By directly remapping source training data into the target domain, we remove the low-level differences between the domains, ensuring that our task model is well-conditioned on target data. We depict here the image-level adaptation as composed of the pixel GAN loss (green), the source cycle loss (red), and the source and target semantic consistency losses (black dashed) – used when needed to prevent label flipping. For clarity the target cycle is omitted. The feature-level adaptation is depicted as the feature GAN loss (orange) and the source task loss (purple).

and class-specific statistics, while mining additional temporal data from street view datasets to learn a static object prior. Zhang et al. (2017) instead perform segmentation adaptation by aligning label distributions both globally and across superpixels in an image.

## 3 CYCLE-CONSISTENT ADVERSARIAL DOMAIN ADAPTION

We consider the problem of unsupervised adaptation, where we are provided source data $X_S$, source labels $Y_S$, and target data $X_T$, but no target labels. The goal is to learn a model $f_T$ that correctly predicts the label for the target data $X_T$.

**Pretrain Source Task Model.** We begin by simply learning a source model $f_S$ that can perform the task on the source data. For $K$-way classification with a cross-entropy loss, this corresponds to

$$\mathcal{L}_{\text{task}}(f_S, X_S, Y_S) = -\mathbb{E}_{(x_s,y_s)\sim(X_S,Y_S)} \sum_{k=1}^{K} \mathbb{1}_{[k=y_s]} \log\left(\sigma(f_S^{(k)}(x_s))\right) \qquad (1)$$

where $\sigma$ denotes the softmax function. However, while the learned model $f_S$ will perform well on the source data, typically domain shift between the source and target domain leads to reduced performance when evaluating on target data.

**Pixel-level Adaptation.** To mitigate the effects of domain shift, we follow previous adversarial adaptation approaches and learn to map samples across domains such that an adversarial discriminator is unable to distinguish the domains. By mapping samples into a common space, we enable our model to learn on source data while still generalizing to target data.

To this end, we introduce a mapping from source to target $G_{S\to T}$ and train it to produce target samples that fool an adversarial discriminator $D_T$. Conversely, the adversarial discriminator attempts to classify the real target data from the source target data. This corresponds to the loss function

$$\mathcal{L}_{\text{GAN}}(G_{S\to T}, D_T, X_T, X_S) = \mathbb{E}_{x_t\sim X_T}\left[\log D_T(x_t)\right] + \mathbb{E}_{x_s\sim X_S}\left[\log(1 - D_T(G_{S\to T}(x_s)))\right] \quad (2)$$

This objective ensures that $G_{S\to T}$, given source samples, produces convincing target samples. In turn, this ability to directly map samples between domains allows us to learn a target model $f_T$ by minimizing $\mathcal{L}_{\text{task}}(f_T, G_{S\to T}(X_S), Y_S)$ (see Figure 2 green portion).

However, while previous approaches that optimized similar objectives have shown effective results, in practice they can often be unstable and prone to failure. Although the GAN loss in Equation 2 ensures that $G_{S\to T}(x_s)$ for some $x_s$ will resemble data drawn from $X_T$, there is no way to guarantee that $G_{S\to T}(x_s)$ preserves the structure or content of the original sample $x_s$.

In order to encourage the source content to be preserved during the conversion process, we impose a cycle-consistency constraint on our adaptation method (Zhu et al., 2017; Yi et al., 2017; Kim et al., 2017) (see Figure 2 red portion). To this end, we introduce another mapping from target to source $G_{T \to S}$ and train it according to the same GAN loss $\mathcal{L}_{\text{GAN}}(G_{T \to S}, D_S, X_S, X_T)$. We then require that mapping a source sample from source to target and back to the source reproduces the original sample, thereby enforcing cycle-consistency. In other words, we want $G_{T \to S}(G_{S \to T}(x_s)) \approx x_s$ and $G_{S \to T}(G_{T \to S}(x_t)) \approx x_t$. This is done by imposing an L1 penalty on the reconstruction error, which is referred to as the *cycle-consistency loss*:

$$\mathcal{L}_{\text{cyc}}(G_{S \to T}, G_{T \to S}, X_S, X_T) = \mathbb{E}_{x_s \sim X_S} \left[ ||G_{T \to S}(G_{S \to T}(x_s)) - x_s||_1 \right] \tag{3}$$
$$+ \mathbb{E}_{x_t \sim X_T} \left[ ||G_{S \to T}(G_{T \to S}(x_t)) - x_t||_1 \right].$$

Additionally, as we have access to source labeled data, we explicitly encourage high semantic consistency before and after image translation. We used the pretrained source task model $f_S$, as a noisy labeler by which we encourage an image to be classified in the same way after translation as it was before translation according to this classifier. Let us define the predicted label from a fixed classifier, $f$, for a given input $X$ as $p(f, X) = \arg \max(f(X))$. Then we can define the semantic consistency before and after image translation as follows:

$$\mathcal{L}_{\text{sem}}(G_{S \to T}, G_{T \to S}, X_S, X_T, f_S) = \mathcal{L}_{\text{task}}(f_S, G_{T \to S}(X_T), p(f_S, X_T)) \tag{4}$$
$$+ \mathcal{L}_{\text{task}}(f_S, G_{S \to T}(X_S), p(f_S, X_S))$$

See Figure 2 black portion. This can be viewed as analogously to content losses in style transfer (Gatys et al., 2016) or in pixel adaptation (Taigman et al., 2017a), where the shared content to preserve is dictated by the source task model $f_S$.

**Feature-level Adaptation.** We have thus far described an adaptation method which combines cycle consistency, semantic consistency, and adversarial objectives to produce a final target model. As a pixel-level method, the adversarial objective consists of a discriminator which distinguishes between two image sets, e.g. transformed source and real target image. Note that we could also consider a feature-level method which discriminates between the features or semantics from two image sets as viewed under a task network. This would amount to an additional feature level GAN loss (see Figure 2 orange portion):

$$\mathcal{L}_{\text{GAN}}(f_T, D_{\text{feat}}, f_S(G_{S \to T}(X_S)), X_T). \tag{5}$$

Taken together, these loss functions form our complete objective:

$$\mathcal{L}_{\text{CyCADA}}(f_T, X_S, X_T, Y_S, G_{S \to T}, G_{T \to S}, D_S, D_T) \tag{6}$$
$$= \mathcal{L}_{\text{task}}(f_T, G_{S \to T}(X_S), Y_S)$$
$$+ \mathcal{L}_{\text{GAN}}(G_{S \to T}, D_T, X_T, X_S) + \mathcal{L}_{\text{GAN}}(G_{T \to S}, D_S, X_S, X_T)$$
$$+ \mathcal{L}_{\text{GAN}}(f_T, D_{\text{feat}}, f_S(G_{S \to T}(X_S)), X_T)$$
$$+ \mathcal{L}_{\text{cyc}}(G_{S \to T}, G_{T \to S}, X_S, X_T) + \mathcal{L}_{\text{sem}}(G_{S \to T}, G_{T \to S}, X_S, X_T, f_S).$$

This ultimately corresponds to solving for a target model $f_T$ according to the optimization problem

$$f_T^* = \arg \min_{f_T} \min_{\substack{G_{S \to T} \\ G_{T \to S}}} \max_{D_S, D_T} \mathcal{L}_{\text{CyCADA}}(f_T, X_S, X_T, Y_S, G_{S \to T}, G_{T \to S}, D_S, D_T). \tag{7}$$

We have introduced a method for unsupervised adaptation which views prior adversarial objectives as operating at the pixel or feature level and generalizes to a method which may benefit from both approaches. In addition, we introduce the combination of cycle-consistency together with semantic transformation constraints to regularize the mapping from one domain to another. We apply CyCADA to both digit classification and to semantic segmentation. We implement $G_S$ and $G_T$ as a pixel-to-pixel convnet, $f_S$ and $f_T$ as a convnet classifier or a Fully-Convolutional Net (FCN), and $D_S$, $D_T$, and $D_{\text{feat}}$ as a convnet with binary outputs.

## 4 EXPERIMENTS

We evaluate CyCADA on several unsupervised adaptation scenarios. We first focus on adaptation for digit classification using the MNIST (LeCun et al., 1998), USPS, and Street View House Numbers

| Model | MNIST → USPS | USPS → MNIST | SVHN → MNIST |
|---|---|---|---|
| Source only | 82.2 ± 0.8 | 69.6 ± 3.8 | 67.1 ± 0.6 |
| DANN (Ganin et al., 2016) | - | - | 73.6 |
| DTN (Taigman et al., 2017a) | - | - | 84.4 |
| CoGAN (Liu & Tuzel, 2016a) | 91.2 | 89.1 | - |
| ADDA (Tzeng et al., 2017) | 89.4 ± 0.2 | 90.1 ± 0.8 | 76.0 ± 1.8 |
| CyCADA pixel only | **95.6 ± 0.2** | 96.4 ± 0.1 | 70.3 ± 0.2 |
| CyCADA pixel+feat | **95.6 ± 0.2** | **96.5 ± 0.1** | **90.4 ± 0.4** |
| Target only | 96.3 ± 0.1 | 99.2 ± 0.1 | 99.2 ± 0.1 |

Table 2: **Unsupervised domain adaptation across digit datasets.** Our model is competitive with or outperforms state-of-the-art models for each shift. For the difficult shift of SVHN to MNIST we also note that feature space adaptation provides additional benefit beyond the pixel-only adaptation.

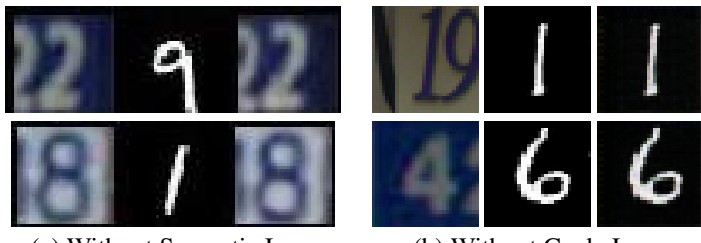

(a) Without Semantic Loss             (b) Without Cycle Loss

Figure 3: **Ablation: Effect of Semantic or Cycle Consistency** Examples of translation failures without the semantic consistency loss. Each triple contains the original SVHN image (*left*), the image translated into MNIST style (*middle*), and the image reconstructed back into SVHN (*right*). (a) Without semantic loss, both the GAN and cycle constraints are satisfied (translated image matches MNIST style and reconstructed image matches original), but the image translated to the target domain lacks the proper semantics. (b) Without cycle loss, the reconstruction is not satisfied and though the semantic consistency leads to some successful semantic translations (*top*) there are still cases of label flipping (*bottom*).

(SVHN) (Netzer et al., 2011) datasets. After which we present results for the task of semantic image segmentation, using the GTA (Richter et al., 2016) and CityScapes (Cordts et al., 2016) datasets, see Appendix A.1.2 for an additional experiment with the SYNTHIA (Ros et al., 2016a) dataset.

## 4.1 DIGIT ADAPTATION

We evaluate our method across the adaptation shifts of USPS to MNIST, MNIST to USPS, and SVHN to MNIST, using the full training sets during learning phases and evaluating on the standard test sets. We report classification accuracy for each shift in Table 2 and find that our method outperforms competing approaches on average. The classifier for our method for all digit shifts uses a variant of the LeNet architecture (see A.1.1 for full implementation details). Note that the recent pixel-da method by Bousmalis et al. (2017b) presents results for only the MNIST to USPS shift and reports 95.9% accuracy, while our method achieves 95.6% accuracy. However, the pixel-da approach cross validates with some labeled data which is not an equivalent evaluation setting.

**Ablation: Pixel vs Feature Level Transfer.** We begin by evaluating the contribution of the pixel space and feature space transfer. We find that in the case of the small domain shifts between USPS and MNIST, the pixel space adaptation by which we train a classifier using images translated using CycleGAN (Zhu et al., 2017), performs very well, outperforming or comparable to prior adaptation approaches. Feature level adaptation offers a small benefit in this case of a small pixel shift. However, for the more difficult shift of SVHN to MNIST, we find that feature level adaptation outperforms the pixel level adaptation, and importantly, both may be combined to produce an overall model which outperforms all competing methods.

**Ablation: No Semantic Consistency.** We experiment without the addition of our semantic consistency loss and find that the standard unsupervised CycleGAN approach diverged when training SVHN to MNIST often suffering from random label flipping. Figure 3(a) demonstrates two examples

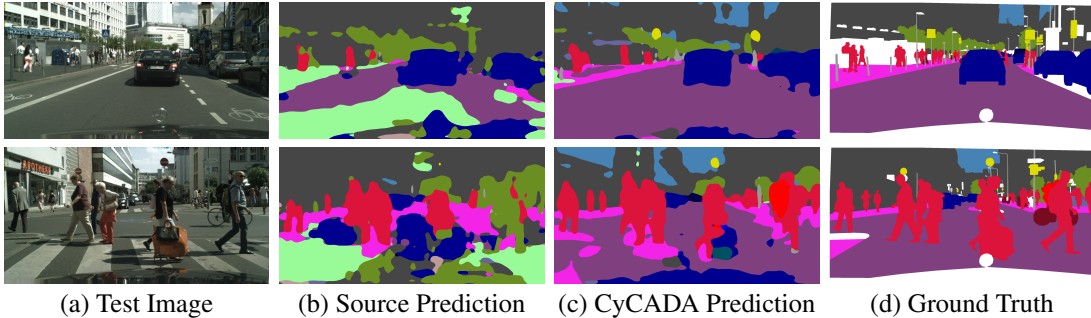

|(a) Test Image | (b) Source Prediction | (c) CyCADA Prediction | (d) Ground Truth |

Figure 4: **GTA5 to CityScapes Semantic Segmentation.** Each test CityScapes image (a) along with the corresponding predictions from the source only model (b) and our CyCADA model (c) are shown and may be compared against the ground truth annotation (d).

where cycle constraints alone fail to have the desired behavior for our end task. An SVHN image is mapped to a convincing MNIST type image and back to a SVHN image with correct semantics. However, the MNIST-like image has mismatched semantics. Our modified version, which uses the source labels to train a weak classification model which can be used to enforce semantic consistency before and after translation, resolves this issue and produces strong performance.

**Ablation: No Cycle Consistency.** We study the result when learning without the cycle consistency loss. First note that there is no reconstruction guarantee in this case, thus in Figure 3(b) we see that the translation back to SVHN fails. In addition, we find that while the semantic loss does encourage correct semantics it relies on the weak source labeler and thus label flipping still occurs (see right image triple).

## 4.2 SEMANTIC SEGMENTATION ADAPTATION

The task is to assign a semantic label to each pixel in the input image, e.g. $road$, $building$, etc. We limit our evaluation to the unsupervised adaptation setting, where labels are only available in the source domain, but we are evaluated solely on our performance in the target domain.

For each experiment, we use three metrics to evaluate performance. Let $n_{ij}$ be the number of pixels of class $i$ predicted as class $j$, let $t_i = \sum_j n_{ij}$ be the total number of pixels of class $i$, and let $N$ be the number of classes. Our three evaluation metrics are, mean intersection-over-union (mIoU), frequency weighted intersection-over-union (fwIoU), and pixel accuracy, which are defined as follows:

$$\text{mIoU} = \frac{1}{N} \cdot \frac{\sum_i n_{ii}}{t_i + \sum_j n_{ji} - n_{ii}}, \text{fwIoU} = \frac{1}{\sum_k t_k} \cdot \frac{\sum_i n_{ii}}{t_i + \sum_j n_{ji} - n_{ii}}, \text{pixel acc.} = \frac{\sum_i n_{ii}}{\sum_i t_i}.$$

Cycle-consistent adversarial adaptation is general and can be applied at any layer of a network. Since optimizing the full CyCADA objective in Equation 6 end-to-end is memory-intensive in practice, we train our model in stages. First, we perform image-space adaptation and map our source data into the target domain. Next, using the adapted source data with the original source labels, we learn a task model that is suited to operating on target data. Finally, we perform another round of adaptation between the adapted source data and the target data in feature-space, using one of the intermediate layers of the task model. Additionally, we do not use the semantic loss for the segmentation experiments as it would require loading generators, discriminators, and an additional semantic segmenter into memory all at once for two images. We did not have the required memory for this at the time of submission, but leave it to future work to deploy model parallelism or experiment with larger GPU memory.

To demonstrate our method's applicability to real-world adaptation scenarios, we also evaluate our model in a challenging synthetic-to-real adaptation setting. For our synthetic source domain, we use the GTA5 dataset (Richter et al., 2016) extracted from the game Grand Theft Auto V, which contains 24966 images. We consider adaptation from GTA5 to the real-world Cityscapes dataset (Cordts et al., 2016), from which we used 19998 images without annotation for training and 500 images for validation. Both of these datasets are evaluated on the same set of 19 classes, allowing for

| | Architecture | road | sidewalk | building | wall | fence | pole | traffic light | traffic sign | vegetation | terrain | sky | person | rider | car | truck | bus | train | motorbike | bicycle | mIoU | fwIoU | Pixel acc. |
|---|---|---|---|---|---|---|---|---|---|---|---|---|---|---|---|---|---|---|---|---|---|---|---|
| **GTA5 → Cityscapes** | | | | | | | | | | | | | | | | | | | | | | | |
| Source only | A | 26.0 | 14.9 | 65.1 | 5.5 | 12.9 | 8.9 | 6.0 | 2.5 | 70.0 | 2.9 | 47.0 | 24.5 | 0.0 | 40.0 | 12.1 | 1.5 | 0.0 | 0.0 | 0.0 | 17.9 | 41.9 | 54.0 |
| FCNs in the wild* | A | 70.4 | 32.4 | 62.1 | 14.9 | 5.4 | 10.9 | 14.2 | 2.7 | 79.2 | 21.3 | 64.6 | 44.1 | 4.2 | 70.4 | 8.0 | 7.3 | 0.0 | 3.5 | 0.0 | 27.1 | — | — |
| CyCADA feat-only | A | **85.6** | 30.7 | 74.7 | 14.4 | 13.0 | 17.6 | 13.7 | 5.8 | 74.6 | 15.8 | **69.9** | 38.2 | 3.5 | 72.3 | 16.0 | 5.0 | 0.1 | 3.6 | 0.0 | 29.2 | 71.5 | 82.5 |
| CyCADA pixel-only | A | 83.5 | **38.3** | 76.4 | 20.6 | **16.5** | 22.2 | **26.2** | 21.9 | 80.4 | 28.7 | 65.7 | 49.4 | 4.2 | 74.6 | 16.0 | 26.6 | 2.0 | 8.0 | 0.0 | 34.8 | 73.1 | 82.8 |
| CyCADA pixel+feat | A | 85.2 | 37.2 | **76.5** | 21.8 | 15.0 | **23.8** | 22.9 | 21.5 | **80.5** | **31.3** | 60.7 | **50.5** | **9.0** | **76.9** | **17.1** | **28.2** | **4.5** | **9.8** | 0.0 | **35.4** | **73.8** | **83.6** |
| Oracle - Target Super | A | 96.4 | 74.5 | 87.1 | 35.3 | 37.8 | 36.4 | 46.9 | 60.1 | 89.0 | 54.3 | 89.8 | 65.6 | 35.9 | 89.4 | 38.6 | 64.1 | 38.6 | 40.5 | 65.1 | 60.3 | 87.6 | 93.1 |
| Source only | B | 42.7 | 26.3 | 51.7 | 5.5 | 6.8 | 13.8 | 23.6 | 6.9 | 75.5 | 11.5 | 36.8 | 49.3 | 0.9 | 46.7 | 3.4 | 5.0 | 0.0 | 5.0 | 1.4 | 21.7 | 47.4 | 62.5 |
| CyCADA feat-only | B | 78.1 | 31.1 | 71.2 | 10.3 | 14.1 | 29.8 | 28.1 | 20.9 | 74.0 | 16.8 | 51.9 | 53.6 | 6.1 | 65.4 | 8.2 | 20.9 | 1.8 | 13.9 | 5.9 | 31.7 | 67.4 | 78.4 |
| CyCADA pixel-only | B | 63.7 | 24.7 | 69.3 | 21.2 | 17.0 | 30.3 | 33.0 | **32.0** | 80.5 | 25.3 | 62.3 | 62.0 | **15.1** | 73.1 | 19.8 | 23.6 | 5.5 | 16.2 | **28.7** | 37.0 | 63.8 | 75.4 |
| CyCADA pixel+feat | B | **79.1** | **33.1** | **77.9** | 23.4 | 17.3 | 32.1 | **33.3** | 31.8 | **81.5** | 26.7 | 69.0 | 62.8 | 14.7 | **74.5** | 20.9 | 25.6 | 6.9 | 18.8 | 20.4 | **39.5** | 72.4 | 82.3 |
| Oracle - Target Super | B | 97.3 | 79.8 | 88.6 | 32.5 | 48.2 | 56.3 | 63.6 | 73.3 | 89.0 | 58.9 | 93.0 | 78.2 | 55.2 | 92.2 | 45.0 | 67.3 | 39.6 | 49.9 | 73.6 | 67.4 | 89.6 | 94.3 |

Table 3: Adaptation between GTA5 and Cityscapes, showing IoU for each class and mean IoU, freq-weighted IoU and pixel accuracy. CyCADA significantly outperforms baselines, nearly closing the gap to the target-trained oracle on pixel accuracy. *FCNs in the wild is by Hoffman et al. (2016). We compare our model using two base semantic segmentation architectures (A) VGG16-FCN8s (Long et al., 2015) base network and (B) DRN-26 (Yu et al., 2017).

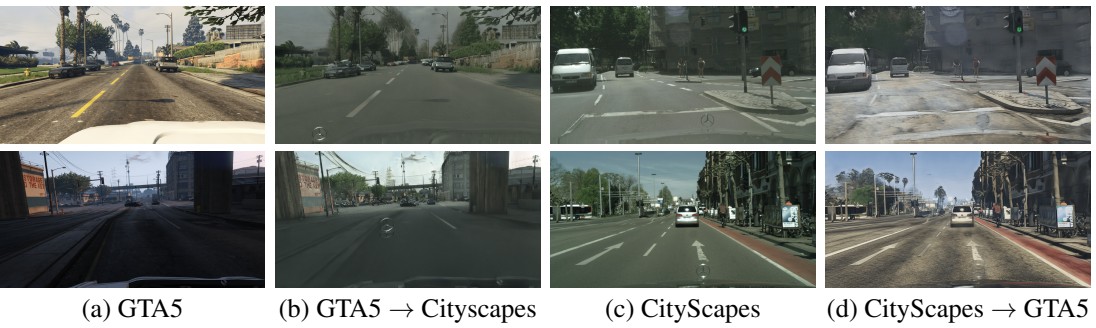

|  (a) GTA5  |  (b) GTA5 → Cityscapes  |  (c) CityScapes  |  (d) CityScapes → GTA5  |

Figure 5: **GTA5 to CityScapes Image Translation.** Example images from the GTA5 (a) and Cityscapes (c) datasets, alongside their image-space conversions to the opposite domain, (b) and (d), respectively. Our model achieves highly realistic domain conversions.

straightforward adaptation between the two domains. For an additional experiment evaluating cross-season adaptation in synthetic environments see the Appendix.

Image-space adaptation also affords us the ability to visually inspect the results of the adaptation method. This is a distinct advantage over opaque feature-space adaptation methods, especially in truly unsupervised settings—without labels, there is no way to empirically evaluate the adapted model, and thus no way to verify that adaptation is improving task performance. Visually confirming that the conversions between source and target images are reasonable, while not a *guarantee* of improved task performance, can serve as a sanity check to ensure that adaptation is not completely diverging. This process is diagrammed in Figure 2. For implementation details please see Appendix A.1.2.

### 4.2.1 SYNTHETIC TO REAL ADAPTATION

To evaluate our method's applicability to real-world adaptation settings, we investigate adaptation from synthetic to real-world imagery. The results of this evaluation are presented in Table 3 with qualitative results shown in Figure 4. Once again, CyCADA achieves state-of-the-art results, recovering approximately 40% of the performance lost to domain shift. CyCADA also improves or maintains performance on all 19 classes. Examination of fwIoU and pixel accuracy as well as individual class IoUs reveals that our method performs well on most of the common classes. Although some classes such as *train* and *bicycle* see little or no improvement, we note that those classes are poorly

represented in the GTA5 data, making recognition very difficult. We compare our model against Shrivastava et al. (2017) for this setting, but found this approach did not converge and resulted in worse performance than the source only model (see Appendix for full details).

We visualize the results of image-space adaptation between GTA5 and Cityscapes in Figure 5. The most obvious difference between the original images and the adapted images is the saturation levels—the GTA5 imagery is much more vivid than the Cityscapes imagery, so adaptation adjusts the colors to compensate. We also observe texture changes, which are perhaps most apparent in the road: in-game, the roads appear rough with many blemishes, but Cityscapes roads tend to be fairly uniform in appearance, so in converting from GTA5 to Cityscapes, our model removes most of the texture. Somewhat amusingly, our model has a tendency to add a hood ornament to the bottom of the image, which, while likely irrelevant to the segmentation task, serves as a further indication that image-space adaptation is producing reasonable results.

## 5 CONCLUSION

We presented a cycle-consistent adversarial domain adaptation method that unifies cycle-consistent adversarial models with adversarial adaptation methods. CyCADA is able to adapt even in the absence of target labels and is broadly applicable at both the pixel-level and in feature space. An image-space adaptation instantiation of CyCADA also provides additional interpretability and serves as a useful way to verify successful adaptation. Finally, we experimentally validated our model on a variety of adaptation tasks: state-of-the-art results in multiple evaluation settings indicate its effectiveness, even on challenging synthetic-to-real tasks.

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

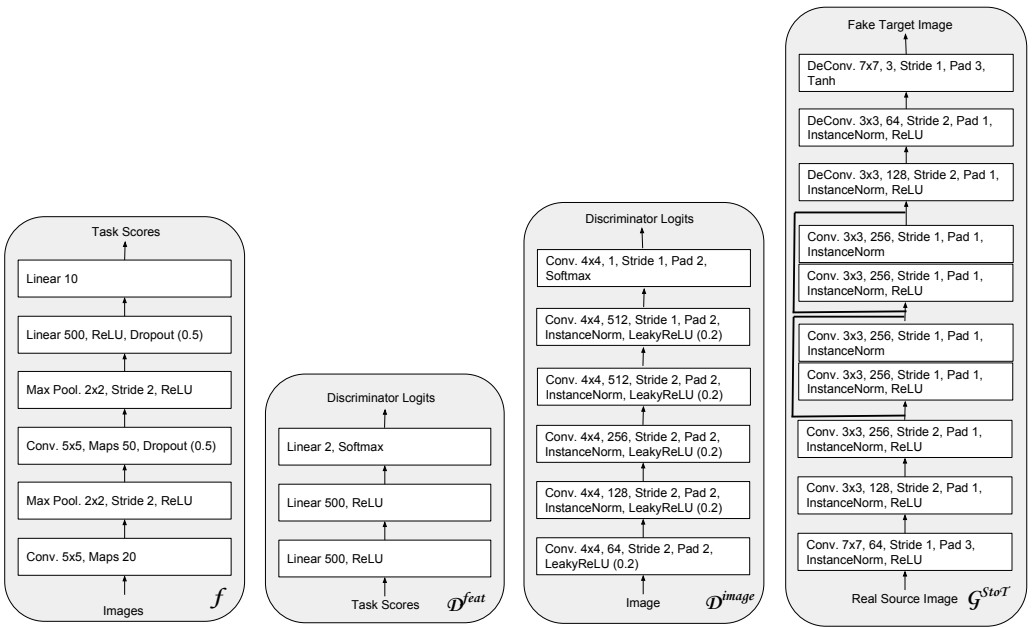

Figure 6: Network architectures used for digit experiments. We show here the task net ($f$), discriminator for feature level adaptation ($D^{feat}$), discriminator for image level adaptation ($D^{image}$), and generator for source to target ($G$) – same network used for target to source.

# A   Appendix

## A.1   Implementation Details

We begin by pretraining the source task model, $f_S$, using the task loss on the labeled source data. Next, we perform pixel-level adaptation using our image space GAN losses together with semantic consistency and cycle consistency losses. This yeilds learned parameters for the image transformations, $G_{S\rightarrow T}$ and $G_{T\rightarrow S}$, image discriminators, $D_S$ and $D_T$, as well as an initial setting of the task model, $f_T$, which is trained using pixel transformed source images and the corresponding source pixel labels. Finally, we perform feature space adpatation in order to update the target semantic model, $f_T$, to have features which are aligned between the source images mapped into target style and the real target images. During this phase, we learn the feature discriminator, $D_{\text{feat}}$ and use this to guide the representation update to $f_T$. In general, our method could also perform phases 2 and 3 simultaneously, but this would require more GPU memory then available at the time of these experiments.

For all feature space adaptation we equally weight the generator and discriminator losses. We only update the generator when the discriminator accuracy is above 60% over the last batch (digits) or last 100 iterations (semantic segmentation) – this reduces the potential for volatile training. If after an epoch (entire pass over dataset) no suitable discriminator is found, the feature adaptation stops, otherwise it continues until max iterations are reached.

### A.1.1   Digit Experiments

For all digit experiments we use a variant of the LeNet architecture as the task net (Figure 6 *left*). Our feature discriminator network consists of 3 fully connected layers (Figure 6 *mid left*). The image discriminator network consists of 6 convolutional layers culminating in a single value per pixel (Figure 6 *mid right*). Finally, to generate one image domain from another we use a multilayer network which consists of convolution layers followed by two residual blocks and then deconvolution layers (Figure 6 *right*). All stages are trained using the Adam optimizer.

**Hyperparameters.** For training the source task net model, we use learning rate 1e-4 and train for 100 epochs over the data with batch size 128. For feature space adaptation we use learning rate 1e-5 and train for max 200 epochs over the data. For pixel space adaptation we train our generators and discriminators with equal weighting on all losses, use batch size 100, learning rate 2e-4 (default from CycleGAN), and trained for 50 epochs. We ran each experiment 4 times and report the average and standard error across the runs.

### A.1.2 SEMANTIC SEGMENTATION

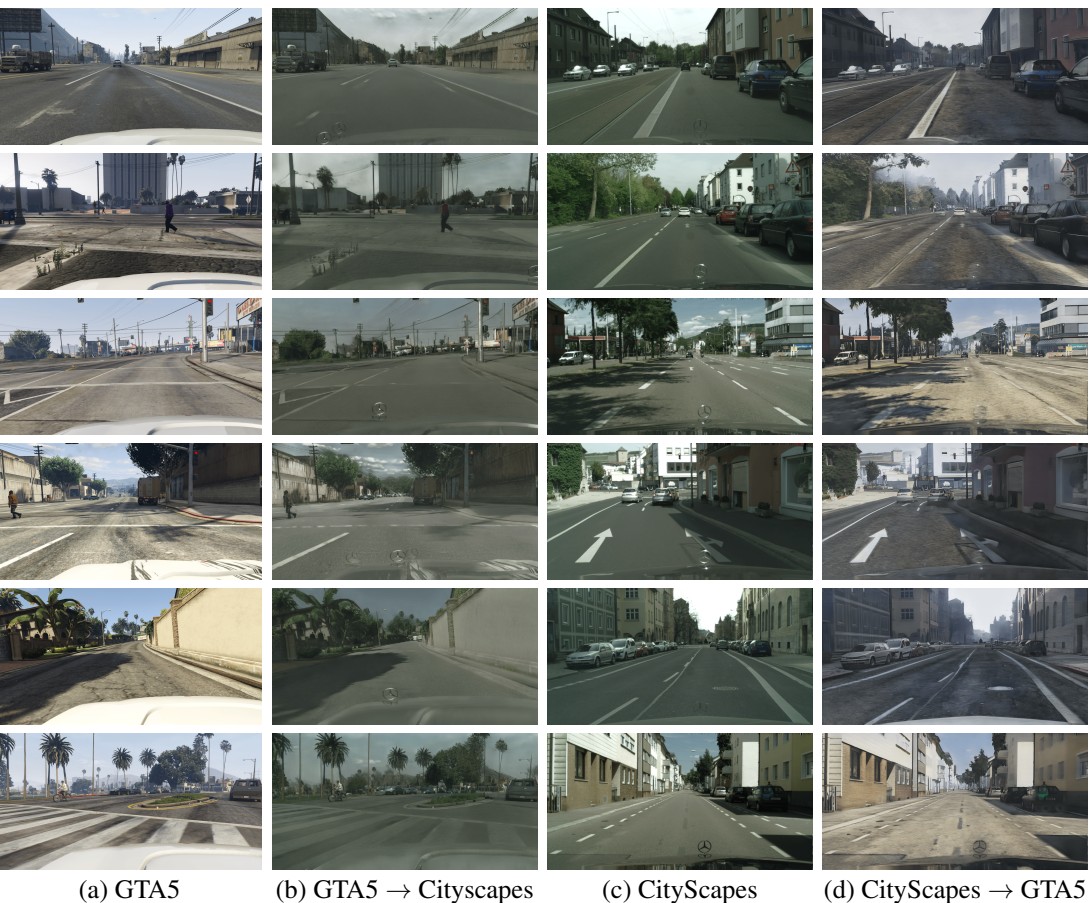

(a) GTA5     (b) GTA5 → Cityscapes     (c) CityScapes     (d) CityScapes → GTA5

Figure 7: **GTA5 to CityScapes Image Translation.** Example images from the GTA5 (a) and Cityscapes (c) datasets, alongside their image-space conversions to the opposite domain, (b) and (d), respectively. Our model achieves highly realistic domain conversions.

We experiment with both the VGG16-FCN8s Long et al. (2015) architecture as well as the DRN-26 Yu et al. (2017) architecture. For FCN8s, we train our source semantic segmentation model for 100k iterations using SGD with learning rate 1e-3 and momentum 0.9. For the DRN-26 architecture, we train our source semantic segmentation model for 115K iterations using SGD with learning rate 1e-3 and momentum 0.9. We use a crop size of 600x600 and a batch size of 8 for this training. For cycle-consistent image level adaptation, we followed the network architecture and hyperparameters of CycleGAN(Zhu et al., 2017). All images were resized to have width of 1024 pixels while keeping the aspect ratio, and the training was performed with randomly cropped patches of size 400 by 400. Also, due to large size of the dataset, we trained only 20 epochs. For feature level adaptation, we train using SGD with momentum, 0.99, and learning rate 1e-5. We weight the representation loss ten times less than the discriminator loss as a convenience since otherwise the discriminator did not learn a suitable model within a single epoch. Then the segmentation model was trained separately using the adapted source images and the ground truth labels of the source data. Due to memory limitations

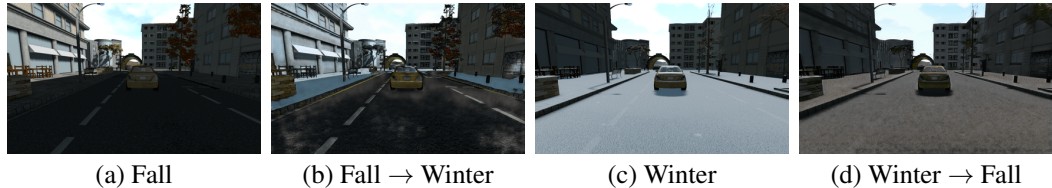

| (a) Fall | (b) Fall → Winter | (c) Winter | (d) Winter → Fall |

Figure 8: **Cross Season Image Translation.** Example image-space conversions for the SYNTHIA seasons adaptation setting. We show real samples from each domain (Fall and Winter) alongside conversions to the opposite domain.

| **SYNTHIA Fall → Winter** | | | | | | | | | | | | | | | |
|---|---|---|---|---|---|---|---|---|---|---|---|---|---|---|---|
| | sky | building | road | sidewalk | fence | vegetation | pole | car | traffic sign | pedestrian | bicycle | lanemarking | traffic light | **mIoU** | **fwIoU** | **Pixel acc.** |
| Source only | 91.7 | 80.6 | 79.7 | 12.1 | 71.8 | 44.2 | 26.1 | 42.8 | 49.0 | 38.7 | 45.1 | 41.3 | 24.5 | 49.8 | 71.7 | 82.3 |
| FCNs in the wild | 92.1 | 86.7 | 91.3 | 20.8 | 72.7 | **52.9** | **46.5** | 64.3 | 50.0 | **59.5** | **54.6** | **57.5** | 26.1 | 59.6 | — | — |
| CyCADA pixel-only | **92.5** | **90.1** | **91.9** | **79.9** | **85.7** | 47.1 | 36.9 | **82.6** | 45.0 | 49.1 | 46.2 | 54.6 | 21.5 | **63.3** | **85.7** | **92.1** |
| Oracle (Train on target) | 93.8 | 92.2 | 94.7 | 90.7 | 90.2 | 64.4 | 38.1 | 88.5 | 55.4 | 51.0 | 52.0 | 68.9 | 37.3 | 70.5 | 89.9 | 94.5 |

Table 4: Adaptation between seasons in the SYNTHIA dataset. We report IoU for each class and mean IoU, freq-weighted IoU and pixel accuracy. Our CyCADA method achieves state-of-the-art performance on average across all categories. *FCNs in the wild is by Hoffman et al. (2016).

we can only include a single source and single target image at a time (crops of size 768x768), this small batch is one of the main reasons for using a high momentum parameter.

### A.1.3 Cross-season adaptation

As an additional semantic segmentation evaluation, we consider the SYNTHIA dataset (Ros et al., 2016a), which contains synthetic renderings of urban scenes. We use the SYNTHIA video sequences, which are rendered across a variety of environments, weather conditions, and lighting conditions. This provides a synthetic testbed for evaluating adaptation techniques. For comparison with previous work, in this work we focus on adaptation between seasons. We use only the front-facing views in the sequences so as to mimic dashcam imagery, and adapt from fall to winter. The subset of the dataset we use contains 13 classes and consists of 10,852 fall images and 7,654 winter images.

We start by exploring the abilities of pixel space adaptation alone (using FCN8s architecture) for the setting of adapting across seasons in synthetic data. For this we use the SYNTHIA dataset and adapt from fall to winter weather conditions. Typically in unsupervised adaptation settings it is difficult to interpret what causes the performance improvement after adaptation. Therefore, we use this setting as an example where we may directly visualize the shift from fall to winter and inspect the intermediate pixel level adaptation result from our algorithm. In Figure 8 we show the result of pixel only adaptation as we generate a winter domain image (b) from a fall domain image (a), and visa versa (c-d). We may clearly see the changes of adding or removing snow. This visually interpretable result matches our expectation of the true shift between these domains and indeed results in favorable final semantic segmentation performance from fall to winter as shown in Table 4. We find that CyCADA achieves state-of-the-art performance on this task with image space adaptation alone, however does not recover full supervised learning performance (train on target). Some example errors includes adding snow to the sidewalks, but not to the road, while in the true winter domain snow appears in both locations. However, even this mistake is interesting as it implies that the model is learning to distinguish road from sidewalk during pixel adaptation, despite the lack of pixel annotations.

Cycle-consistent adversarial adaptation achieves state-of-the-art adaptation performance. We see that under the fwIoU and pixel accuracy metrics, CyCADA approaches oracle performance, falling short by only a few points, despite being entirely unsupervised. This indicates that CyCADA is extremely effective at correcting the most common classes in the dataset. This conclusion is supported by

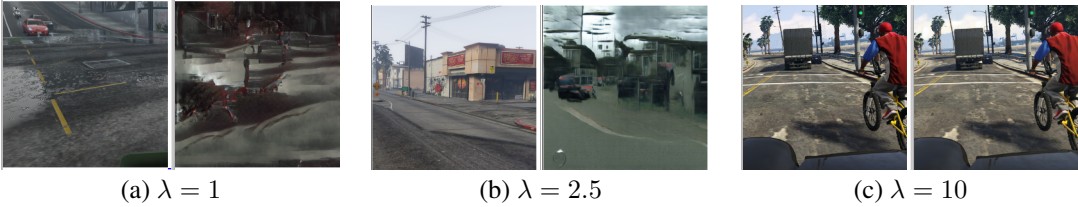

(a) $\lambda = 1$        (b) $\lambda = 2.5$        (c) $\lambda = 10$

Figure 9: Image transformation results from Shrivastava et al. (2017) applied to GTA to CityScapes transformation. We demonstrate results using three different settings for $\lambda$.

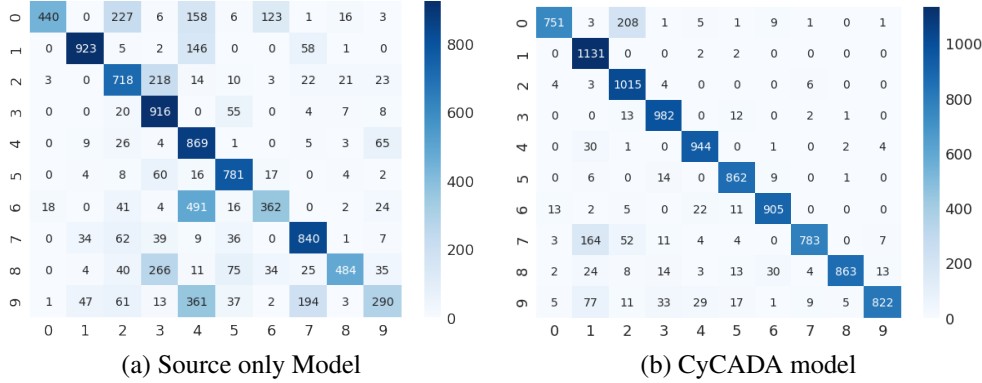

(a) Source only Model        (b) CyCADA model

Figure 10: Confusion matrices for SVHN → MNIST experiment.

inspection of the individual classes in Table 4, where we see the largest improvement on common classes such as *road* and *sidewalk*.

### A.2 COMPARISON TO SHRIVASTAVA ET AL. (2017) FOR SEMANTIC SEGMENTATION

We illustrate the performance of a recent pixel level adaptation approach proposed by Shrivastava et al. (2017) on our semantic segmentation data – GTA to Cityscapes. These images are significantly larger and more complex than those shown in the experiments in the original paper. We show image to image translation results under three different settings of the model hyperparameter, $\lambda$, which controls the tradeoff between the reconstruction loss and the visual style loss. When $\lambda = 10$ (Figure 9 *right*), the resulting image converges to a near replica of the original image, thus preserving content but lacking the correct target style. When $\lambda = 1$ or $\lambda = 2.5$ (Figure 9 *left*), the results lack any consistent semantics making it difficult to perceive the style of the transformed image. Thus, the resulting performance for this model is 11.6 mIoU for FCN8s with VGG, well below the performance of the corresponding source model of 17.9 mIoU.

### A.3 EXPERIMENT ANALYSIS

To understand the types of mistakes which are improved upon and those which still persist after adaptation, we present the confusion matrices before and after our approach for the digit experiment of SVHN to MNIST (Figure 10). Before adaptation we see common confusions are 0s with 2s, 4s, and 7s. 6 with 4, 8 with 3, and 9 with 4. After adaptation all errors are reduced, but we still find that 7s are confused with 1s and 0s with 2s. These errors make some sense as with hand written digits, these digits sometimes resemble one another. It remains an open question to produce a model which may overcome these types of errors between highly similar classes.

