# OpenReview forum: "CyCADA: Cycle-Consistent Adversarial Domain Adaptation"
_ICLR.cc/2018/Conference — Reject_

### Official Review · AnonReviewer1 · 2017-11-26
**Novelty incremental, results encouraging, writing could be improved**

**Rating:** 5
**Confidence:** 5

**Review:**

This paper proposed a domain adaptation approach by extending the CycleGAN with 1) task specific loss functions and 2) loss imposed over both pixels and features. Experiments on digit recognition and semantic segmentation verify the effectiveness of the proposed method.

Strengths:
+ It is a natural and intuitive application of CycleGAN to domain adaptation.
+ Some of the implementation techniques may be useful for the future use of CycleGAN or GAN in other applications, e.g., the regularization over both pixels and features, etc.
+ The experimental results are superior over the past.
+ The translated images in Figure 6 are amazing. Could the authors show more examples and include some failure cases (if any)?

Weaknesses:
- The presentation of the paper could be improved. I do not think I can reproduce the experimental results after reading the paper more than twice. Many details are missing and some parts are confusing or even misleading.  As below, I highlight a few points and the authors are referred to the comments by Cedric Nugteren for more suggestions.

-- Equation (4) is incorrect.
-- In the introduction and approach sections, it reads like a big deal to adapt on both the pixel and feature levels. However, the experiments fail to show that these two levels of adaptation are complementary to each other. Either the introduction is a little misleading or the experiments are insufficient.
-- What does the “image-space adaptation” mean?
-- There are three fairly sophisticated training stages in Section 4.2. However, the description of the three stages are extremely short and ambiguous.
-- What are exactly the network architectures used in the experiments?

- The technical contribution seems like only marginal innovative.
- The experiments adapting from MNIST to SVHN would be really interesting, given that the MNIST source domain is not as visually rich as the SVHN target. Have the authors conducted the corresponding experiments? How are the results?

Summary:
The proposed method is a natural application of CycleGAN to domain adaptation. The technical contribution is only marginal. The results on semantic segmentation are encouraging and may motivate more research along this direction. It is unfortunate that the paper writing leaves many parts of the paper unclear.

=========================================
Post rebuttal:

The rebuttal addresses my first set of questions. The revised paper describes more experiment details, corrects equation (4), and clarifies some points about the results.

This paper applies the cycle consistent GAN to domain adaptation. I still think the technical contribution is only marginally innovative. Nonetheless, I do not weigh this point too much given that the experiments are very extensive.

The rebuttal does not answer my last question. It would be interesting to see what happens to adapt from MNIST to SVHN, the latter of which contains more complicated background than the former.

---

> ### Author Response · Authors · 2017-12-28
> **Revision: Text clarifications and implementation details in appendix**
>
> Thank you for your comments. We have made a number of modifications to our manuscript based on your feedback. First, thank you for noticing the error in Equation 4. We have updated it to accurately reflect our description and implementation (our new figure 2 should also clarify its use). We have modified the explanation of image/pixel space adaptation vs feature space adaptation within the main method description and provided headers to guide the reader. We have also added an appendix with an implementation section specifying the network architectures and describing the training procedures. We will release our code, data and models upon publication. We have also followed many of the suggestions from Cedric Nugteren as you have pointed out (please see our response there for the detailed list of changes).
>
> We would like to clarify that our results show that independently pixel space and feature space adaptation offer performance improvement over no adaptation across all experiments. When combined they provide anywhere from equivalent (as in USPS<->MNIST) to marginal improvement (GTA->CityScapes), to *significantly* better performance (SVHN->MNIST) than either approach alone.  Thus, we propose using both components together.

---

### Official Review · AnonReviewer2 · 2017-11-29
**Great problem and idea, but without adequate experiments that show that cycle-consistency is the cause of the improvement**

**Rating:** 5
**Confidence:** 5

**Review:**

This paper essentially uses CycleGANs for Domain Adaptation. My biggest concern is that it doesn't adequately compare to similar papers that perform adaptation at the pixel level (eg. Shrivastava et al-'Learning from Simulated and Unsupervised Images through Adversarial Training' and Bousmalis et al - 'Unsupervised Pixel-level Domain Adaptation with GANs', two similar papers published in CVPR 2017 -the first one was even a best paper- and available on arXiv since December 2016-before CycleGANs). I believe the authors should have at least done an ablation study to see if the cycle-consistency loss truly makes a difference on top of these works-that would be the biggest selling point of this paper. The experimental section had many experiments, which is great. However I think for semantic segmentation it would be very interesting to see whether using the adapted synthetic GTA5 samples would improve the SOTA on Cityscapes. It wouldn't be unsupervised domain adaptation, but it would be very impactful. Finally I'm not sure the oracle (train on target) mIoU on Table 2 is SOTA, and I believe the proposed model's performance is really far from SOTA.

Pros:
* CycleGANs for domain adaptation! Great idea!
* I really like the work on semantic segmentation, I think this is a very important direction

Cons:
* I don't think Domain separation networks is a pixel-level transformation-that's a feature-level transformation, you probably mean to use Bousmalis et al. 2017. Also Shrivastava et al is missing from the image-level papers.
* the authors claim that Bousmalis et al, Liu & Tuzel and Shrivastava et al ahve only been shown to work for small image sizes. There's a recent work by Bousmalis et al. (Using Simulation and Domain Adaptation to Improve Efficiency of Deep Robotic Grasping) that shows these methods working well (w/o cycle-consistency) for settings similar to semantic segmentation at a relatively high resolution. Also it was mentioned that these methods do not necessarily preserve content, when pixel-da explicitly accounts for that with a task loss (identical to the semantic loss used in this submission)
* The authors talk about the content similarity loss on the foreground in Bousmalis et al. 2017, but they could compare to this method w/o using the content similarity or using a different content similarity tailored to the semantic segmentation tasks, which would be trivial.
* Math seems wrong in (4) and (6). (4) should be probably have a minus instead of a plus. (6) has an argmin of a min, not sure what is being optimized here. In fact, I'm not sure if eg you use the gradients of f_T for training the generators?
* The authors mention that the pixel-da approach cross validates with some labeled data. Although I agree that is not an ideal validation, I'm not sure if it's equivalent or not the authors' validation setting, as they don't describe what that is.
* The authors present the semantic loss as novel, however this is the task loss proposed by the pixel-da paper.
* I didn't understand what pixel-only and feat-only meant in tables 2, 3, 4. I couldn't find an explanation in captions or in text


=====
Post rebuttal comments:
Thanks for adding content in response to my comments. The cycle ablation is still a sticky point for me, and I'm still left not sure if cycle-consistency really offers an improvement. Although I applaud your offering examples of failures for when there's no cycle-consistency, these are circumstantial and not quantitative.  The reader is still left wondering why and when is the cycle-consistency loss is appropriate. As this is the main novelty, I believe this should be in the forefront of the experimental evaluation.

---

> ### Author Response · Authors · 2017-12-28
> **New semantic segmentation experiments, comparison to Shrivastava et al, cycle ablation, text improvements**
>
> Thank you for your comments. We have included new experiments and text edits per your suggestion.
>
> Higher performing semantic segmentation models
> ======================================
> First, we added a new experiment for GTA->CityScapes adaptation with a newer semantic segmentation model. Again, we found that for this experiment, feature space adaptation alone provided a large improvement (21 mIoU -> 31 mIoU), pixel adaptation alone resulted in a substantial improvement (21->37 mIoU) and finally, combining feature space with pixel space adaptation provided the largest performance (21->39).
>
> Cycle Ablation
> ===========
> We added a new ablation experiment to the SVHN->MNIST setting where the cycle loss is removed while the semantic loss remains. This version was still susceptible to label flipping and understandably failed at the task of reconstruction (see Figure 3b).
>
> Comparison to other Pixel Level Approaches
> ==================================
> We ran Shrivastava et al (see Appendix A.2) in the GTA->CityScapes scenario and found that the model was not able to accurately capture the transfer problem, resulting in performance below the original source model.
>
> We added a citation to the new Bousmalis et al. (2017a) paper on robotic grasping (pg 1 Introduction). Those images are indeed higher resolution than the prior work, but they still do not match the resolution of the dashcam driving images and have significantly lower variation and complexity. In general, optimizing pixel transfer methods with high resolution images remains a challenging problem. Our approach provides one solution by which additional regularization through the pixel cycle loss encourages transfer. We would like to clarify that the comment we made about prior pixel level approaches which “may not necessarily preserve content” was intended as a potential criticism of pixel based approaches in general, not specifically about Bousmalis et al. (2017b). In fact, in the related work section we explicitly mention that Bousmalis et al. (2017b) uses a content similarity loss on the foreground mask. This is a privileged version of our semantic consistency loss as it requires a known foreground mask on target data. We do not claim to be the first to introduce the use the a task classifier to preserve content. Instead we introduce a model which does pixel transfer through a cycle loss for low level preservation and a semantic loss for preserving semantics in a large domain shift scenario (when all pixels must change significantly).
>
> Text Edits
> =======
> Thank you for noticing the error in Equation (4). We have updated the text to accurately reflect our description and implementation. In addition, we have added semantic consistency to our new Figure 2 to clarify the use of this objective.
>
> Equation (6) defines the full CyCADA objective and Equation (7) presents the optimization problem.
>
> Appendix A.1 describes architectures, training procedures, and implementation details needed to reproduce our experiments.
>
> We have revised the method section to clarify the pixel vs feature level transfer which is ablated in the experiments section. In addition the new Figure 2 should offer further clarity.

---

### Official Review · AnonReviewer3 · 2017-11-30
**This paper extends the previous work on CycleGAN by coupling it with adversarial adaptation approaches. The extension includes a new feature and semantic loss in the overall objective of the CycleGAN. While this extension is straightforward, it is novel. The experimental validation is extensive and clearly shows the benefits of the proposed extension.**

**Rating:** 9
**Confidence:** 5

**Review:**

This paper proposes  a natural extension of the CycleGAN approach. This is achieved by leveraging the feature and semantic losses to achieve a more realistic image reconstruction. The experiments show that including these additional losses is critical for improving the models performance.  The paper is very well written and technical details are well described and motivated. It would be good to identify the cases where the model fails and comment on those. For instance, what if the source data cannot be well reconstructed from adapted target data? What are the bounds of the domain discrepancy in this case?

---

> ### Author Response · Authors · 2017-12-28
> **New revision and analysis for digit experiments**
>
> Thank you for your positive feedback and suggestion to study the errors from the model. We have included a section to our appendix illustrating the confusion matrices for the largest domain shift of our digit experiments -- SVHN -> MNIST. In this case we find certain error types are resolved after adaptation while others still remain. Confusion between visually similar classes, such as 1s and 7s, is difficult to resolve without target labels.
>
> In addition, we have included additional experiments and made updates to further clarify details within our manuscript based on the suggestions from the other reviewers.

---

### Public Comment · ~Cedric_Nugteren1 · 2017-11-10
**Some remarks on the presentation of the work**


This is a not a review of the work, but just a comment with some suggestions to improve the presentation of the work. Currently there are some things unclear and inconsistent in the presentation; I believe improving this can make the contributions of the paper a lot clearer. Here are some comments (not in any particular order):

* Figure 2 (the diagram with images, networks, and losses) is really helpful. However, it would help if the symbols used in the paper (Xs, Xt, Yt, Lgan, Ft, Lcyc, etc.) are added to make it easier to map the equations to the figure. Also, it would be good to extend the figure with the second cycle loss. I understand that that takes extra space, but it might be worth it. Furthermore, it would be good to picture the missing parts as well (Lsem, Fs) for completeness. Finally, perhaps explicitly adding all networks would help clarifying the overall structure (Ds, Dt are missing now).

* In equation 1 (task-loss) it would be clarifying to put large square brackets around the "-sum()" term. Now the equation could be read as "expectation minus the sum of ..." whereas it should read as "expectation of the negated sum of ...".

* Equation 4 has a typo. The left-hand side contains a Gt->s component but it is not on the right-hand side. It would be furthermore helpful to clarify what the two individual components in this equation represent.

* It would be good to make explicit early on that the source model fs has fixed weights throughout the domain adaptation training. Is this also the case in related work?

* At first it is unclear how the "pixel" and "feature" approaches discussed in the experiment section map to the explanation in section 3 and figure 2. It would be good to clarify this in section 3 and perhaps in a second version of figure 2? There are some unclarities here:
  - Are all loss components trained for the feature case?
  - How are the features obtained? Using the task-model? What if these features are not useful for the target domain (e.g. color information not present in MNIST features but might be useful for SVHN)?
  - Which networks are shared between the pixel and feature approaches?
  - How are the two losses optimized - one after each other? Interleaved? Jointly?

* There seem to be some assumptions on the domain change with respect to the fact that the source labels Ys do not need to be transformed to accommodate changed made on the input data Xs by the transformation Gs->t (e.g. no translation, warping, etc.). It would be nice if this is mentioned explicit and perhaps discussed (is Gs->t constrained in such a way?).

* The first paragraph under the section "Implementation details" doesn't seem to be an implementation detail at all, but rather a property of the approach.

* The network architecture used (FCN) is quite old in terms of semantic segmentation (2015). It would be interesting to see how this affects your final accuracy. Is this why the only comparison is against "FCNs in the wild", perhaps they use the same architecture? If not, how much of your improvement is related to the architecture change and how much related to the method?

* Table 3 contains some results which are better than the oracle (pole, pedestrian, bicycle). Although possible, it would be good to mention this explicitly to make sure this is not a typo.

---

> ### Public Comment · ~Lei_Tai1 · 2017-11-17
> **Feature and pixel loss**
>
> This is a public comment. I agree with the comments of Cedric Nugteren especially for the feature and pixel loss.
>
> In addition:
> 1. In the last paragraph of section 3, it said CyCADA can be viewed as CycleGan augmented with an additional task loss, which I think should be the semantic loss here? But in Table I, CyCADA also covers feature loss and CycleGAN doesn't.
> From the Equation 5, the loss should be presented as the feature or the pixel loss explicitly. Otherwise, the training in stages from section 4.2 really makes me confused.
>
> 2. In right side of Equation 4, L_task(f_s, X_s, p(fs, Xs)) is not a loss function if f_s is pre-trained. It just outputs a constant.

---

> > ### Author Response · Authors · 2017-12-28
> > **Revision addresses comments.**
> >
> > Thank you for your interest and suggestions. We have addressed Cedric’s comments above. In addition, we have made changes to our method section to clarify the distinction between pixel and feature space adaptation. The new Appendix 6.1 discusses the training procedure which indicates which components are updated in each phase. Equation (4) has been fixed.

---

> ### Author Response · Authors · 2017-12-28
> **Clarifications in new revision**
>
> Thank you for your suggestions on how to improve the presentation of our algorithm. We have incorporated them into our revised manuscript. Changes are specified below.
>
> * Figure 2: we include a new version of this figure with semantic consistency and feature level transfer together with explicit discriminator blocks.
>
> * Moved (-) outside the expectation in Equation (1)
>
> * Fixed Equation (4)
>
> *Made explicit that the source model is pretrained and fixed. Also see implementation details in the Appendix which reinforces this.
>
> *Pixel and Feature adaptation are clarified in the method section as well as in the appendix implementation section.
>
> * Indeed, we do assume that the label space remains unchanged before and after transfer. In fact, that is exactly what the semantic consistency loss enforces.
> New semantic segmentation results with the DRN-26 architecture results in higher performance overall. Our findings remain the same.

---

### Decision · Program_Chairs · 2018-01-29
**ICLR 2018 Conference Acceptance Decision**

**Decision:**

Reject

**Comment:**

I concur with two of the reviewers: the work is somewhat incremental in terms of technical novelty (it's effectively CycleGANs for domain adaptation with a couple of effective tricks) and the need/advantage of the cycle consistency loss is not demonstrated sufficiently. The only solid ablation evidence seems to the the SVHN-->MNIST experiment from post-submission; I would personally like to see this kind of empirical proof extended much further (the fact that Shrivastava et al.'s method doesn't work well on GTA-->Cityscapes is not itself proof that cycle consistency is needed). With more empirical evidence I can see this paper being a good candidate for a computer vision conference like CVPR or ICCV.